# Pairwise Proximal Policy Optimization: Language Model Alignment with Comparative RL

**Tianhao Wu**[1]*   **Banghua Zhu**[1]   **Ruoyu Zhang**[1]   **Zhaojin Wen**[1]
**Kannan Ramchandran**[1]   **Jiantao Jiao**[1]
[1]University of California, Berkeley

## Abstract

LLMs may exhibit harmful behavior without aligning with human values. The dominant approach for steering LLMs towards beneficial behavior is Reinforcement Learning with Human Feedback (RLHF). This involves training a reward model with a human-labeled ranking dataset and fine-tuning the LLM with the reward signal using RL. Despite the fact that the reward is learned from comparing different responses, the RL stage doesn't involve direct comparisons. This inconsistency between reward learning and reinforcement learning stages exacerbates RL's instability. An example would be that the well adopted RL optimizer, Proximal Policy Optimization (PPO), could perform different gradient updates even for batches with identical human preference information. To address this, we propose a new framework, reinforcement learning with comparative feedback, and a simple policy gradient algorithm, Pairwise Proximal Policy Optimization (P3O), that learns to improve from direct comparison. Theoretically, P3O has the nice property of being invariant with any reward functions that contain identical preference information, while doesn't require learning a value function. Empirical evaluations demonstrate that P3O can align with human preferences better than existing methods. This suggest that comparative RL is strong candidate for aligning LLM with preference data.

## 1 Introduction

Large Language Models (LLMs) have made remarkable progress, profoundly influencing the AI community (Chowdhery et al., 2022; Brown et al., 2020; Touvron et al., 2023; Bubeck et al., 2023). However, due to the reliance on massive corpora of internet data, which encompasses a high portion of low-quality data, LLMs are likely to express unintended behavior. These include fabricating facts, generating biased or toxic text, and even harmful content to humans (Perez et al., 2022; Ganguli et al., 2022). Consequently, it is crucial to align LLMs with human values, *e.g.*, helpful, honest, harmless (Bai et al., 2022a).

A leading method in AI Alignment for Large Language Models (LLMs), known as Reinforcement Learning from Human Feedback (RLHF), involves learning a reward function and fine-tuning the model with this reward feedback using reinforcement learning (RL) (Ziegler et al., 2019; Ouyang et al., 2022). Specifically, a reward model is trained to rank candidate responses to align with the human-labeled ground-truth. As for RL, Proximal Policy Optimization (PPO) is widely adopted as the default optimizer (Schulman et al., 2017). PPO alternate between generate new responses and adjust the likelihood toward responses with higher reward. Despite its acclaimed efficiency, we identify the inconsistency between these two stages:

**Inconsistency Between Reward Learning and RL.** Although rewards are derived by ranking responses according to human judgments, the RL phase does not incorporate comparisons between generated samples. This leads to a scenario where the reward signal can be highly variable, often being lower for more challenging prompts and higher for simpler ones.

---

*Corresponding to thw@berkeley.edu

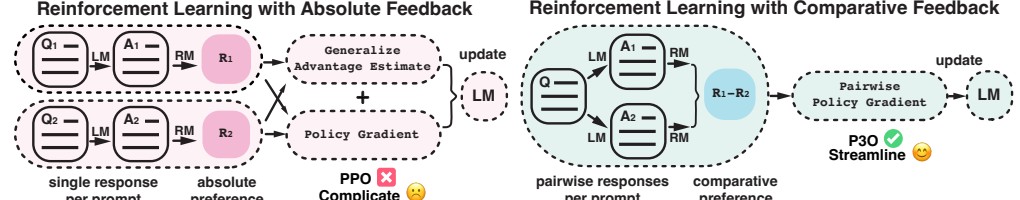

Figure 1: The figure on the left illustrates the prevalent method for fine-tuning LMs using RL, which relies on **Absolute Feedback**. In this paradigm, algorithms like PPO has to learn a $V$ function, which capture not only the valuable relative preference information, but also less part, which is the scale of the reward for a given prompt. Contrastingly, the figure on the right presents RL paradigm that improve from direct comparison. Our algorithm generate a pair of responses per prompt, leveraging only the **Comparative Feedback** - derived from the difference in reward - for policy gradient updates. This method obviates the need for additional critic learning and intricate components like Generalized Advantage Estimation (Schulman et al., 2015b).

When employing conventional algorithms like Proximal Policy Optimization (PPO) to optimize against such a noisy reward, it becomes likely for PPO to reduce the likelihood of a quality response to a difficult prompt due to the absence of comparisons. This issue is further compounded by PPO's sensitivity to various factors including reward normalization, scaling, clipping, KL control, advantage normalization, and critic initialization (Zheng et al., 2023; Engstrom et al., 2020), contributing to its fragility. As a consequence, we observe that PPO's reward experiences a decline during a warm-up period, and the training is sensitive to random seed. From a formal perspective, we argue this inconsistency by noting that the reward training objective, the Bradley-Terry Loss, is invariant to a constant shift (Yan et al., 2022), whereas PPO is not. This implies that even if two reward models contain identical human preference information, their optimization via PPO could lead to disparate results. (subsection 4.3)

In this paper, we provide new insights to address the inconsistency:

• We define an equivalent relationship for reward functions trained from human preferences. We identify that the widely adopted reward training loss, Bradley-Terry Loss, is invariant under this equivalent relationship, while PPO is not. As a result, PPO may be less efficient at learning the reward.

• We introduce Pairwise Proximal Policy Optimization (P3O), under the framework of Reinforcement Learning with Comparative Feedback (Figure 1). P3O learns by comparing pairs of responses, avoiding learning critic functions, advantage estimation and various normalization techniques (Zheng et al., 2023). Empirical evaluations show that P3O consistently outperforms PPO and Direct Preference Optimization (DPO) in terms of GPT-4 Evaluation.

## 2  Related Work

Significant efforts have been made towards aligning LLMs with human values. These alignment strategies broadly fall into two categories: offline training and online training.

Offline training typically involve a static dataset, and doesn't require additional evaluations or generations. For instance, Thoppilan et al. (2022); Gunasekar et al. (2023) use instruction fine-tuning to update the model on a high quality dataset tailored to a specific downstream task of interest. Snell et al. (2022) proposed to employ offline $Q$ Learning to learn an add-on term for decoding. While Rafailov et al. (2023) introduced DPO, an offline approach that can directly align LM with human preference data, drawing from the closed-form solution of the Contextual Bandit with KL control problem. There are also methods like PRO (Song et al., 2023) and RRHF (Yuan et al., 2023) that fine-tune the model based on ranking of the

rewards. Liu et al. (2024) unified several pairwise objectives under the LiPO framework and establish connection with Learning-to-Rank Literature.

Our work is categorized under online training, which consist of a loop of generating new responses from the updated policy, evaluating them with the reward model and updating the policy. The current dominant approach RLHF relies on online RL methods such as PPO (Schulman et al., 2017), A2C (Mnih et al., 2016) or their variants (Ramamurthy et al., 2022; Zhu et al., 2023c). There are also few methods that deviate from this standard. For instance, ReST (Gulcehre et al., 2023) use offline RL in the policy improvement phase. RAFT (Dong et al., 2023a) iteratively fine-tune the policy on the responses generated by the Best-of-N policy. Another paradigm parallel to RLHF is Reinforcement Learning with AI Feedback (RLAIF) (Zhu et al., 2023a;b; Bai et al., 2022b; Lee et al., 2023; Yuan et al., 2024), which aim for using AI to improve AI. For example, Wu et al. (2024) proposed Meta-Rewarding LMs, which significantly improve the instruction-following and judging ability without relying on human data.

Outside of the context of language, Contextual dueling bandit (Dudík et al., 2015; Yue et al., 2012) use preferences or rankings of actions to adjust the policy, rather than rewards. Similarly, PbRL (Xu et al., 2020; Jain et al., 2013; Busa-Fekete et al., 2014; Christiano et al., 2017; Sadigh et al., 2017; Kupcsik et al., 2018) learn from binary preferences generated by some unknown scoring function.

## 2.1 Necessity of RL in LLM Alignment

The necessity of reinforcement learning (RL) for aligning large language models (LLMs) has been a topic of much debate. There are alternative approaches, such as Direct Policy Optimization (DPO), which is a simpler method that utilizes a pre-collected offline dataset of preferences. On the other hand, RL involves a more intricate process of interacting with the language model in real-time. This process includes generating new responses, evaluating them using a reward model, and then updating the language model based on the rewards.

Despite the complexity and the challenge of hyperparameter optimization inherent to online RL methods, recent studies have demonstrated that, with careful hyperparameter tuning, online RL can produce results that are much stronger than non-RL methods Zhu et al. (2023a); Lambert & Calandra (2023); Dong et al. (2023b); Xu et al. (2024). However, conclusively addressing the necessity of RL for LLM alignment would require more sophisticated tooling and evaluations, exceeds the scope of this paper.

## 3 Preliminaries

We briefly reviewing the RLHF pipeline in (Ziegler et al., 2019).

- **SFT Phase (Supervised Fine-Tuning):** This stage start with a pre-trained LM, and then fine-tuned with supervised learning (typically maximum likelihood loss) on a high quality dataset for the downstream task of interest. Outcome of this stage is denoted as $\pi^{\text{SFT}}$.

- **Reward Learning Phase.** In the second phase the SFT model is prompted with prompts $x$ to produce pairs of answers $y_1, y_2 \sim \pi^{\text{SFT}}(y|x)$. The responses pairs are then presented to human labelers who express preferences for one answer, denoted as $y_w \succ y_l | x$, where $y_w$ is the one favored by the labeler and $y_l$ is the one less favored. Under these preferences is the inaccessible latent reward model $r^*(y, x)$. According to the Bradley-Terry (Bradley & Terry, 1952) model, the human preference distribution $p^*$ can be expressed as:

$$p^*(y_1 \succ y_2 | x) = \frac{1}{1 + \exp\left(r^*(y_2|x) - r^*(y_1|x)\right)}$$

Assuming the access to a dataset $\{(x^i, y_w^i, y_l^i)\}_{i=1}^N$ sampled from $p^*$. We parameterize the reward as $r_\phi$ and estimate it via maximum log-likelihood:

$$\sum_{i=1}^N \frac{1}{N} \log \sigma\left(r_\phi(y_w^i|x^i) - r_\phi(y_l^i|x^i)\right) \tag{1}$$

where $\sigma$ is the sigmoid function. $r_\phi$ is initialized with $\pi^{\text{SFT}}$ augmented by additional linear layers on top. Constraints like $\mathbb{E}\left[r(\boldsymbol{y}|\boldsymbol{x})\right] = 0$ might be incorporated to lower the variance.

- **RL Fine-Tuning Phase.** Prior work formulate the optimization problem as maximizing:

$$\mathop{\mathbb{E}}_{\boldsymbol{x}\sim\mathcal{D},\boldsymbol{y}\sim\pi_\theta}\left[r_\phi(\boldsymbol{y}|\boldsymbol{x}) - \beta D_{\text{KL}}(\pi_\theta(\cdot|\boldsymbol{x})\|\pi^{\text{SFT}}(\cdot|\boldsymbol{x}))\right] \tag{2}$$

The $\beta D_{\text{KL}}(\pi_\theta(\cdot|\boldsymbol{x})\|\pi^{\text{SFT}}(\cdot|\boldsymbol{x}))$ term is used to regulate the deviation from the SFT model, it is important to prevent the model from completely forget the world knowledge acquired in the pre-training stage. The standard approach is to directly employ PPO (Schulman et al., 2017; Ouyang et al., 2022) to optimize the modified reward $r_\phi(\boldsymbol{y}|\boldsymbol{x}) - \beta\left(\log \pi_\theta(\boldsymbol{y}|\boldsymbol{x}) - \log \pi^{\text{SFT}}(\boldsymbol{y}|\boldsymbol{x})\right)$.

# 4 Algorithm

## 4.1 Proximal Pairwise Policy Optimization (P3O)

To derive P3O, we start from Vanilla Policy Gradient (VPG, Pseudo 2) (Sutton et al., 1999; Schulman et al., 2017; Wu et al., 2022). For clarity, we'll focus on the bandit setting, though it can be extended to contextual bandits as in Theorem 4.1.

In the bandit setting, assume we are updating a parameterized policy $\pi_\theta$ with actions denoted as $\boldsymbol{y}$. The VPG aims for estimating the following formula with samples:

$$\nabla\mathcal{L}^{\text{VPG}} = \mathop{\mathbb{E}}_{\boldsymbol{y}\sim\pi_\theta}(r(\boldsymbol{y}) - b)\nabla\log\pi_\theta(\boldsymbol{y}) \tag{3}$$

where $b$ is a baseline used for variance reduction. A common choice for the baseline is the mean reward, which gives:

$$\begin{aligned}\nabla\mathcal{L}^{\text{VPG}} &= \mathop{\mathbb{E}}_{\boldsymbol{y}_1\sim\pi_\theta}(r(\boldsymbol{y}_1) - \mathop{\mathbb{E}}_{\boldsymbol{y}_2\sim\pi_\theta}r(\boldsymbol{y}_2))\nabla\log\pi_\theta(\boldsymbol{y}_1) \\ &= \mathop{\mathbb{E}}_{\boldsymbol{y}_1,\boldsymbol{y}_2\sim\pi_\theta}(r(\boldsymbol{y}_1) - r(\boldsymbol{y}_2))\nabla\log\pi_\theta(\boldsymbol{y}_1)\end{aligned} \tag{4}$$

Equation 4 highlights the reliance on the relative difference between rewards. Symmetrizing for $\boldsymbol{y}_1, \boldsymbol{y}_2$ yield:

$$\nabla\mathcal{L}^{\text{VPG}} = \mathop{\mathbb{E}}_{\boldsymbol{y}_1,\boldsymbol{y}_2\sim\pi_\theta}(r(\boldsymbol{y}_1) - r(\boldsymbol{y}_2))\nabla\left(\log\frac{\pi_\theta(\boldsymbol{y}_1)}{\pi_\theta(\boldsymbol{y}_2)}\right)/2$$

Its immediate generalization to contextual bandit is the following:

**Theorem 4.1 (Pairwise Policy Gradient)** *For any prompt $\boldsymbol{x}$, the policy gradient can be expressed as $\nabla\mathcal{L}^{\text{VPG}} = \mathop{\mathbb{E}}_{\boldsymbol{x}\sim\mathcal{D}}\nabla\mathcal{L}^{\text{PPG}}(\boldsymbol{x})$, where $\nabla\mathcal{L}^{\text{PPG}}(\boldsymbol{x})$ can be expressed as:*

$$\mathop{\mathbb{E}}_{\boldsymbol{y}_1,\boldsymbol{y}_2\sim\pi_\theta}(r(\boldsymbol{y}_1|\boldsymbol{x}) - r(\boldsymbol{y}_2|\boldsymbol{x}))\nabla\left(\log\frac{\pi_\theta(\boldsymbol{y}_1|\boldsymbol{x})}{\pi_\theta(\boldsymbol{y}_2|\boldsymbol{x})}\right)/2$$

To estimate the policy gradient with finite samples, considering that the replay buffer is collected using the previous policy $\pi_{\theta_{\text{old}}}$, we further utilize importance sampling to correct the bias. The following theorem provides an unbiased estimation of the pairwise policy gradient:

**Theorem 4.2 (Estimate PPG with Importance Sampling)** *For replay-buffer $\mathcal{D}_k = \{\tau^i = (\boldsymbol{x}^i, \boldsymbol{y}_1^i, \boldsymbol{y}_2^i, r_1^i, r_2^i)\}_{i=1}^n$ collected using $\pi_{\theta_{\text{old}}}$. The following is an unbiased estimation of pairwise policy gradient:*

$$\frac{1}{2n}\sum_{i=1}^n(r_1^i - r_2^i)\frac{\pi_\theta(\boldsymbol{y}_1^i|\boldsymbol{x}^i)}{\pi_{\theta_{\text{old}}}(\boldsymbol{y}_2^i|\boldsymbol{x}^i)}\frac{\pi_\theta(\boldsymbol{y}_2^i|\boldsymbol{x}^i)}{\pi_{\theta_{\text{old}}}(\boldsymbol{y}_2^i|\boldsymbol{x}^i)}\cdot\nabla\left(\log\frac{\pi_\theta(\boldsymbol{y}_1^i|\boldsymbol{x}^i)}{\pi_\theta(\boldsymbol{y}_2^i|\boldsymbol{x}^i)}\right)/2$$

We further employ clipping to stabilize the policy updates, for more details refer to subsection B.2.

---

**Algorithm 1** Pairwise Proximal Policy Optimization (P3O)

---

1: **Initialization:** Initialize policy from the SFT model with parameters $\theta_0$
2: **for** $k = 0, 1, 2 \cdots$ **do**
3:      Sampling $n$ prompts $\{x^i\}_{i=1}^n$ from a prompt dataset. Collect pairwise responses for each prompt by sampling from the latest policy $\pi_{\theta_k}$:

$$y_1^i, y_2^i \sim \pi_{\theta_k}(\cdot | x^i)$$

4:      Score all the responses with a reward model $r_{\text{model}}$, and aggregate the reward with KL divergence:

$$r(y|x) = r_{\text{model}}(y|x) - \beta D_{\text{KL}}\left(\pi_{\theta_k}(\cdot|x) \| \pi_{\theta_0}(\cdot|x)\right)$$

5:      Estimate policy gradient on the scored replay-buffer $\mathcal{D}_k = \{\tau^i = (x^i, y_1^i, y_2^i, r_1^i, r_2^i)\}_{i=1}^n$ via (subsection B.2):

$$\hat{g}_k = \nabla_\theta \mathcal{L}_{\text{joi}}^{\text{P3O}}(\mathcal{D}_k)$$

6:      Update the $\theta_k$ via gradient descent and yield $\theta_{k+1}$.
7: **end for**

---

## 4.2 Relationship with PPO and DPO

**Comparison with PPO:** Although PPO and P3O both fall into the online RL framework, they differ in the way they perform policy update: PPO update based on an estimated advantage, while P3O update based on direct comparison of two responses. Consider a simplified version of PPO applied to contextual bandit:

$$\mathcal{L}_{\text{no clip}}^{\text{PPO}} = - \mathbb{E}_{y \sim \pi_{\theta_{\text{old}}}(\cdot|x)} (r(y|x) - V_\phi(x)) \frac{\pi_\theta(y|x)}{\pi_{\theta_{\text{old}}}(y|x)}$$

Where $V_\phi(x)$ is a proxy to the ground truth value function $V^{\pi_{\theta_{\text{old}}}} = \mathbb{E}_{y \sim \pi_{\theta_{\text{old}}}} r(y|x)$, usually learnt via an additional regression loss. In contrast, P3O doesn't require learning the value function, this significantly reduce engineering efforts. P3O employ an additional sample $y_2$ to estimate the gradient unbiasedly, and update the policy based on the comparison $r_1 - r_2$.

**Comparison with DPO:** The gradient of DPO's objective function $\nabla \mathcal{L}^{\text{DPO}}(x, y_w, y_l)$ takes the following form:

$$\beta \sigma \left( \beta \log \frac{\pi_\theta(y_l|x)}{\pi^{\text{SFT}}(y_l|x)} - \beta \log \frac{\pi_\theta(y_w|x)}{\pi^{\text{SFT}}(y_w|x)} \right) \cdot \nabla \left( \log \frac{\pi_\theta(y_w|x)}{\pi_\theta(y_l|x)} \right) / 2$$

The direction of the gradient resembles that of our formulation in Theorem 4.1. However, the weight coefficients are different. The core difference of DPO and P3O is that P3O is an online RL algorithm while DPO is not. P3O learns to trials and errors by alternating between generate and update from human feedback while DPO is applied to a fixed dataset. We empirically observe that DPO falls short on KL-control (Figure 4 and Figure 2) compared to P3O, we hypothesis that this is because DPO aligns the policy towards the goal policy while doesn't directly consider the reward of the intermediate policies. Unlike P3O, which applies policy gradient based on the idea of strict policy improvement for every gradient update (Schulman et al., 2015a), DPO aligns the policy via an alternate "distance", where the intermediary steps are not guaranteed to maximize the KL-Reward trade-off. We note that P3O combines the benefits of PPO and DPO, offering guaranteed policy improvement akin to policy gradient.

## 4.3 Reward Equivalence

We formally define the concept of reward equivalence (Definition 4.3), similar to the scale invariance defined in (Yan et al., 2022). We show that BTL is invariant under this equivalent

relationship in lemma 4.4. We then discuss why it leads to a desirable property named invariance (Definition 4.5) that we want RL algorithms to satisfy. In the end, we present our main theorem (Theorem 4.6) which shows that PPO does not satisfy this property, contributing to its instability.

**Definition 4.3 (Reward Equivalence)** *Two reward functions $r(\boldsymbol{y}|\boldsymbol{x})$ and $r'(\boldsymbol{y}|\boldsymbol{x})$ are termed equivalent, denoted as $r \sim r'$, if and only if there exist a function $\delta(\boldsymbol{x})$ depend solely on the prompt $\boldsymbol{x}$, such that for every prompt and response pair $(\boldsymbol{x}, \boldsymbol{y})$,*

$$r(\boldsymbol{y}|\boldsymbol{x}) - r'(\boldsymbol{y}|\boldsymbol{x}) = \delta(\boldsymbol{x})$$

*The equivalent class associated with reward $r$ is represented as $[r]$.*

Note that comparative losses such as Bradley-Terry loss and Plackett-Luce loss, is unaffected by a shift in the prompt's reward as in definition 4.3. This observation leads to the following Lemma:

**Lemma 4.4 (Invariance of BTL)** *For two reward functions that satisfy $r \sim r'$, they both yield identical loss for any response pairs (or K responses) under the Bradley-Terry Loss (or Plackett-Luce Loss).*

Lemma 4.4 underscores that the only information we can learn from the preference data is the reward difference of two responses to the same prompt. This implies that direct comparison of responses stemming from different prompts should be avoided. This is because we can craft an arbitrary function denoted as $\delta$ and replace $\hat{r}$ with the identical $\hat{r} + \delta$, while flipping the sign of $\hat{r}(\boldsymbol{y}|\boldsymbol{x}) - \hat{r}'(\boldsymbol{y}'|\boldsymbol{x}')$. As a result, an ideal algorithm should focus only on the relevant information within the reward function, filtering out the noise represented by $\delta$. This leads the following definition:

**Definition 4.5 (Invariance)** *An algorithm is said to be **invariant** with respect to the equivalent relation "$\sim$", if for any two equivalent reward functions $r \sim r'$ and a fixed set of prompt and response pairs, the algorithm perform identical updates to the policy.*

To illustrate definition 4.5, assume that we have two equivalent reward functions $\hat{r}$ and $\hat{r}' = \hat{r} + \delta$. Notably, even when initialized with the same random seed, PPO can result in distinct updates for an identical batch. This behavior can be attributed to PPO's reliance on learning a $V$ function to estimate advantage. In the simplest scenario, where the advantage is estimated via one-step TD ($\mathsf{Adv}(\boldsymbol{y}|\boldsymbol{x}) = r(\boldsymbol{y}|\boldsymbol{x}) - V(\boldsymbol{x})$, corresponding to $\lambda_{\mathrm{GAE}} = 0$) and $\boldsymbol{y}$ is a single token, we should expect the advantage function to stay unchanged. However, following the derivation

$$\mathsf{Adv}_{\hat{r}}(\boldsymbol{y}|\boldsymbol{x}) = \mathsf{Adv}_{\hat{r}'}(\boldsymbol{y}|\boldsymbol{x})$$
$$\iff \hat{r}(\boldsymbol{y}|\boldsymbol{x}) - V_{\hat{r}}(\boldsymbol{x}) = \hat{r}'(\boldsymbol{y}|\boldsymbol{x}) - V_{\hat{r}'}(\boldsymbol{x})$$
$$\iff V_{\hat{r}'}(\boldsymbol{x}) - V_{\hat{r}}(\boldsymbol{x}) = \delta(\boldsymbol{x})$$

We can see that even though $\hat{r}$ and $\hat{r}'$ are equivalent, they yield different updates for $V$ function. This give rise to our main theorem (full proof in subsection C.2):

**Theorem 4.6 (Non-invariance of PPO)** *P3O is invariant with respect to "$\sim$". In contrast, PPO is not, given the same initialization of $V$.*

## 5 Experiments

We conduct experiments on two widely-adopted RLHF tasks, summarization and question-answering, and we find that P3O achieves better performance in terms of both KL-Reward trade-off and quality of generation, against several strong baselines. Refer to Appendix D for detailed hyper parameter selection process.

**Tasks.** We explore two different open-ended text generation tasks, *i.e.* **summarization** and **question-answering**. For both tasks, algorithms are given a reward model pre-trained from

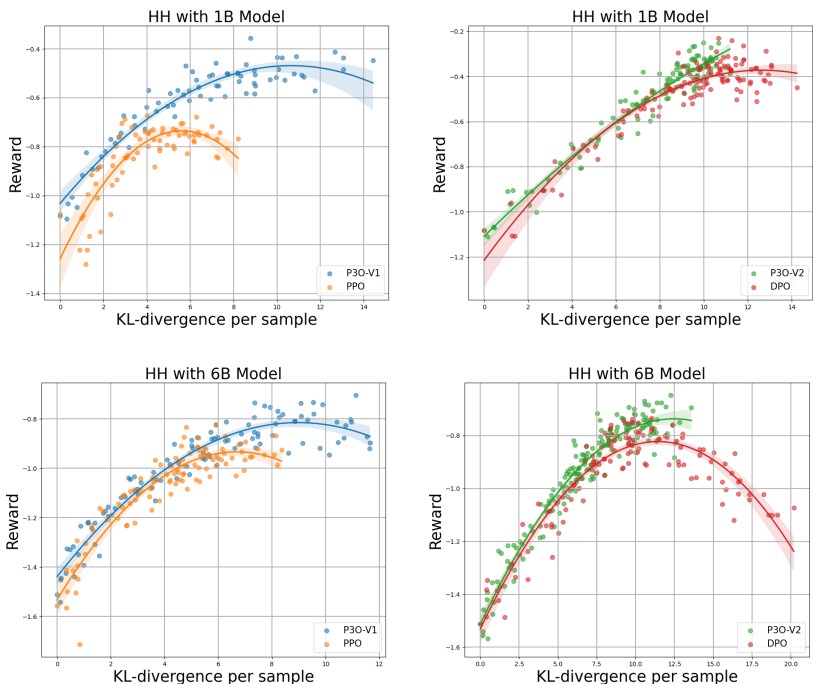

Figure 2: KL-Reward frontier for HH: $x$-axis and $y$-axis represents $D_{\text{KL}}(\pi_\theta \| \pi^{\text{SFT}})$ and the reward respectively. Each point represent an average of results over 280 test prompts and calculated every 500 gradient updates. **Left** two figure compare P3O-V1 and PPO with varying base model sizes; **Right** two figures compare P3O-V2 and online-DPO. Results showing that P3O can not only achieve higher reward but also yield better KL control.

a dataset of preference $\mathcal{D} = \{\boldsymbol{x}^{(i)}, \boldsymbol{y}_w^{(i)}, \boldsymbol{y}_l^{(i)}\}$, and the goal is to obtain a policy $\pi(\boldsymbol{y}|\boldsymbol{x})$ that can generate high-quality response $\boldsymbol{y}$ given prompt $\boldsymbol{x}$. In summarization, we use the **TL;DR** (too long; didn't read) dataset (Völske et al., 2017), where $\boldsymbol{x}$ is a forum post from Reddit, and $\boldsymbol{y}$ is a corresponding summary. We use a 6B SFT model `CarperAI/openai_summarize_tldr_sft` as the initial policy and `EleutherAI/gpt-j-6b` as the reward model. In question-answering, $\boldsymbol{x}$ is a human query, which may come from diverse topics, and the policy should learn to produce an engaging and helpful response $\boldsymbol{y}$. Following prior work, we use the Anthropic Helpful and Harmless (**HH**) dataset (Bai et al., 2022a). We fine-tune two policies of sizes $\{1\text{B},6\text{B}\}$, `Dahoas/pythia-1B-static-sft` and `Dahoas/pythia-6B-static-sft`. Both models have gone through supervised fine-tuning with labeled prompt-response pairs, similar to the protocol in Ouyang et al. (2022) and Ramamurthy et al. (2022). For the reward model, we use the 6B model `Dahoas/gptj-rm-static` trained from the same dataset based on `EleutherAI/gpt-j-6b` as a proxy of human preference.

**Methods.** We compare two versions of P3O, **P3O-V1** and **P3O-V2**, which represent clipping separately and jointly respectively, with several effective and representative approaches for LLM alignment. We start with the **SFT** policy trained by token-wise supervised fine-tuning. It hasn't gone through further alignment; Every other method uses the SFT model as initialization. For RL algorithms[1], we consider the dominant approach **PPO** (Schulman et al., 2017; Ouyang et al., 2022) with reward specified in Equation 2. We follow the implementation of `trlx` (Castricato et al., 2023). Besides, we also consider the newly proposed **DPO** (Rafailov et al., 2023), a method that directly optimizes the policy towards the closed-form solution of the KL-constrained reward maximization. Although DPO is proposed as an offline

---

[1]Among methods directly involving RL, we note that PPO and a modified version A2C (Mnih et al., 2016; Lee et al., 2023) are the only two current online RL methods for LLM alignment. However, there is no strong evidence showing the supremacy of A2C over PPO, so we choose PPO as our baseline.

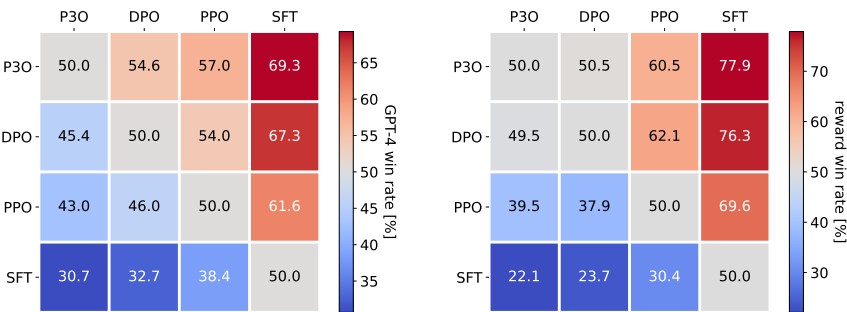

Figure 3: Head-to-head comparisons between {P3O, DPO, PPO, SFT}. **Left** figure displays the win rate as evaluated by GPT-4. **Right** figure presents the win rate based on comparison of the proxy reward. Despite the high correlation between the figures, we found that the reward win rate must be adjusted according to the KL to align with the GPT-4 win rate.

alignment method, we notice that we can make it online with the help of a proxy reward function. (More details can be found in subsection B.3)

**Evaluations.** Deviating too much from the reference policy (*e.g.* SFT model) would lead the online policy to cut corners of the reward model and produce incoherent continuations, as pointed out by previous works (Ziegler et al., 2019). Gao et al. (2023) studied the scaling law of reward over-optimization in a synthetic setup, where labels are supplied by a "gold standard" reward model. They empirically find out the golden reward can be approximated by a simple function form involving the square-root KL-divergence from the reference policy. Therefore, it is important to balance the trade-off between the KL-divergence and asymptotic reward, and we measure the effectiveness of each algorithm by its frontier of achieved reward and KL-divergence from the reference policy (**KL-Reward Frontier**). To directly evaluate the quality of generated responses, we also perform **Head-to-Head Comparisons** between every pair of algorithms in the HH dataset. We use two metrics for evaluation: (1) **Reward**, the optimized target during online RL, (2) **GPT-4**, as a faithful proxy for human evaluation of response helpfulness. For the latter metric, we shall point out that previous studies show that LLMs can be better automated evaluators than existing metrics (Chen et al., 2023), and GPT-4 judgments correlate strongly with humans, with human agreement with GPT-4 typically similar or higher than inter-human annotator agreement (Rafailov et al., 2023). Additional details can be found in Appendix D.

## 5.1 KL-Reward Frontier

We conduct experiments on both TL;DR and HH datasets to evaluate the efficacy of the alignment algorithms in optimizing reward while restricting policy deviation from the reference. Figure 4 and Figure 2 demonstrate the KL-Reward frontier for TL;DR and HH respectively. Each point represents the average evaluation over test prompts at every 500-step interval. The $x$-axis represents the average sequence-level KL-divergence $D_{\mathrm{KL}}(\pi_\theta \| \pi^{\mathrm{SFT}})$, whereas the $y$-axis stands for the average reward given by the proxy reward model. For summarization task, we find that P3O-V1 can reach a slightly higher reward than P3O-V2, while with a worse KL-Reward trade-off. Consequently, only P3O-V2 is included in Figure 4 for comparison. We find that P3O-V2 is able to produce almost

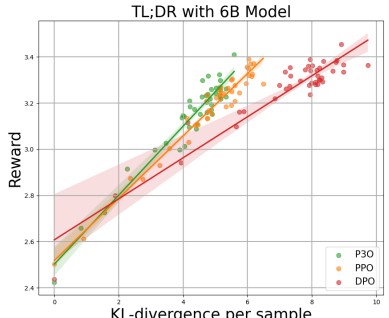

Figure 4: KL-Reward Frontier for TL;DR: The $x$-axis represent $D_{\mathrm{KL}}(\pi_\theta \| \pi^{\mathrm{SFT}})$, $y$-axis represent the reward evaluated by the reward model, both averaged over 200 test prompts and evaluate every 500 gradient steps. We find that a simple linear function fit the curve well. P3O have the best KL-Reward trade-off among the three.

the same highest reward whilst maintaining superior KL efficiency. DPO, despite its faster convergence, exhibits a 25% higher KL-divergence than P3O-V2 under the same reward. For the question-answering task, P3O-V1 and P3O-V2 have strictly dominant frontiers than PPO and DPO respectively in both model sizes, shown by Figure 2. Empirical findings establish P3O's superior trade-off between KL and Reward over other baselines, delivering a substantial higher reward in the range of 0.1-0.3.

## 5.2 Head-to-Head Comparisons

Table 1: Statistics for the checkpoints we used in GPT-4 evaluation: PPO and SFT tends to generate long responses while P3O and DPO generate shorter responses. Moreover, P3O achieves nearly the same reward with DPO while incurring much less KL.

|  | **P3O** | **DPO** | **PPO** | **SFT** |
|---|---|---|---|---|
| Reward ↑ | $-0.302$ | $-0.298$ | $-0.613$ | $-1.195$ |
| KL (sample) ↓ | 9.83 | 12.01 | 7.02 | 0 |
| KL (token) ↓ | 0.12 | 0.14 | 0.06 | 0 |
| Token num | 80.46 | 88.84 | 109.03 | 112.70 |

We further conduct head-to-head comparisons between each algorithm pair among P3O, DPO, PPO and SFT. Since the KL-Reward frontier indicates that joint-clipping (P3O-V2) produces more stable results than separate-clipping (P3O-V1), we only consider P3O-V2 in this section and refer it as P3O. We sample completions from different policies[2] on the test set of the HH dataset at default temperature 1.0, and we compute the average pairwise win rate using (1) **reward** and (2) **GPT-4** as evaluators. Previous studies (Chen et al., 2023; Rafailov et al., 2023) have shown that GPT-4 is a faithful proxy for human preference and is widely adopted for comparisons. The prompt used for evaluation is presented in subsection D.2. Figure 3 presents the comprehensive pairwise comparison results, both via proxy reward and GPT-4. The average KL-divergence and reward ranking of these models is DPO > P3O > PPO > SFT. Although DPO marginally surpasses P3O in reward, it has a considerably higher KL-divergence (Table 1), which may be detrimental to the quality of generation. As a result, DPO has a reward win rate 49.5% against P3O, but only 45.4% as evaluated by GPT-4. Compared with other methods, P3O exhibits a GPT-4 win rate of 57.0% against PPO and 69.3% against SFT. This result is consistent with our findings from the KL-Reward frontier section, affirming that P3O could better align with human preference than previous baselines.

## 6 Conclusion & Future Works

This work presents new insights into aligning large language models with human preferences via reinforcement learning. We introduced the Reinforcement Learning with Comparative Feedback framework, which unifies the core principles of reward modeling and RL fine-tuning. Our empirical evidence compellingly supports the notion that comparative RL, by leveraging direct comparisons, presents a more effective approach for alignment than traditional non-comparative RL methods. Within this framework, we developed P3O, which utilizes pairwise comparisons to perform policy updates. Our empirical assessments have shown that P3O not only surpasses previous methodologies in achieving a better balance on the KL-Reward frontier but also demonstrates superior performance in GPT-4 win-rate comparisons. P3O encapsulates the benefits of policy gradient techniques while simplifying both the algorithmic construction and function approximation.

Looking ahead, several intriguing questions arise for future exploration. Firstly, we aim to understand the impacts of reward over-optimization on trajectory-based RL algorithms and token-based RL algorithms. Secondly, we are interested in whether we can generalize the policy gradient algorithm to accommodate more than two ranked responses, potentially enabling a better trade-off between human effort and AI alignment. Finally, we wish to explore the benefits of applying P3O in contexts beyond training language models with human feedback. We eagerly anticipate investigating these questions in our future work.

---

[2]We select checkpoints with the highest reward for generation.

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

# A  Appendix

# B Algorithms

## B.1 Pseudocodes

---
**Algorithm 2** Vanilla Policy Gradient

---
1: **Initialization:** Initialize policy parameters $\theta_0$ and value function parameters $\phi_0$
2: **for** $k = 0, 1, 2 \cdots$ **do**
3:   Collect trajectories $\mathcal{D}_k = \{\tau_i\}$ by running policy $\pi_{\theta_k}$ starting from a batch of prompts and generate single trajectory from each prompt.
4:   Compute token-wise rewards contain both token-wise KL and preference reward as in Equation 2. And then rewards-to-go $\hat{R}_t$.
5:   Estimate advantage estimates $\widehat{\mathsf{Adv}}_t$ via GAE or other methods.
6:   Estimate policy gradient via:

$$\hat{g}_k = \frac{1}{|\mathcal{D}_k|} \sum_{\tau \in \mathcal{D}_k} \sum_{k=0}^{T} \widehat{\mathsf{Adv}}_t \nabla_\theta \log \pi_\theta(a_t|s_t)$$

7:   Apply gradient updates to $\theta_k$ using gradient descent.
8:   Fit value function by regression on mean-squared error via gradient descent:

$$\phi_{k+1} = \arg\min_\phi \frac{1}{|\mathcal{D}_k|T} \sum_{\tau \in \mathcal{D}_k} \sum_{t=0}^{T} (V_\phi(s_t) - \hat{R}_t)^2$$

9: **end for**

---

We present the pseudocode for both the Vanilla Policy Gradient (VPG) and our proposed algorithm P3O. While both algorithms follow the similar procedure of collecting trajectories and leveraging these trajectories to estimate the gradient, there are key differences: Our method collect pairwise trajectories and compute trajectory-wise rewards. This approach eliminates the need for estimating the value function $V$ and bypasses the requirement of estimating the advantage function using Generalized Advantage Estimation (GAE). Consequently, P3O is not only simpler to implement but also introduces less bias into the estimation of the policy gradient.

## B.2 P3O: combine PPG with clipping

**Clipping Separately (Version 1):** For $\{i, j\} = \{1, 2\}$,

$$\mathcal{L}_i^{\text{P3O}}(x) = \mathop{\mathbb{E}}_{y_1, y_2 \sim \pi_{\theta_{\text{old}}}} \mathsf{sg}\left( (r(y_i|x) - r(y_j|x)) \frac{\pi_\theta(y_j|x)}{\pi_{\theta_{\text{old}}}(y_j|x)} \right) \frac{\pi_\theta(y_i|x)}{\pi_{\theta_{\text{old}}}(y_i|x)}$$

$$\mathcal{L}_{i,\text{clip}}^{\text{P3O}}(x) = \mathop{\mathbb{E}}_{y_1, y_2 \sim \pi_{\theta_{\text{old}}}} \mathsf{sg}\left( (r(y_i|x) - r(y_j|x)) \frac{\pi_\theta(y_j|x)}{\pi_{\theta_{\text{old}}}(y_j|x)} \right) \mathtt{clip}(\frac{\pi_\theta(y_i|x)}{\pi_{\theta_{\text{old}}}(y_i|x)}, 1 - \epsilon, 1 + \epsilon)$$

$$\mathcal{L}_{\text{sep}}^{\text{P3O}} = \mathop{\mathbb{E}}_{x \sim \mathcal{D}} \left[ \min(\mathcal{L}_1^{\text{P3O}}(x), \mathcal{L}_{1,\text{clip}}^{\text{P3O}}(x)) + \min(\mathcal{L}_2^{\text{P3O}}(x), \mathcal{L}_{2,\text{clip}}^{\text{P3O}}(x)) \right] / 2$$

**Clipping Jointly (Version 2):**

$$\mathcal{L}^{\text{P3O}}(x) = \mathop{\mathbb{E}}_{y_1, y_2 \sim \pi_{\theta_{\text{old}}}} \mathsf{sg}\left( (r(y_1|x) - r(y_2|x)) \frac{\pi_\theta(y_1|x)}{\pi_{\theta_{\text{old}}}(y_1|x)} \frac{\pi_\theta(y_2|x)}{\pi_{\theta_{\text{old}}}(y_2|x)} \right) \log \frac{\pi_\theta(y_1|x)}{\pi_\theta(y_2|x)}$$

$$\mathcal{L}_{\text{clip}}^{\text{P3O}}(x) = \mathop{\mathbb{E}}_{y_1, y_2 \sim \pi_{\theta_{\text{old}}}} \mathsf{sg}\left( (r(y_1|x) - r(y_2|x)) \frac{\pi_\theta(y_1|x)}{\pi_{\theta_{\text{old}}}(y_1|x)} \frac{\pi_\theta(y_2|x)}{\pi_{\theta_{\text{old}}}(y_2|x)} \right)$$
$$\times \mathtt{clip}\left( \log \frac{\pi_\theta(y_1|x)}{\pi_\theta(y_2|x)}, \log \frac{\pi_{\theta_{\text{old}}}(y_1|x)}{\pi_{\theta_{\text{old}}}(y_2|x)} - \epsilon, \log \frac{\pi_{\theta_{\text{old}}}(y_1|x)}{\pi_{\theta_{\text{old}}}(y_2|x)} + \epsilon \right)$$

$$\mathcal{L}_{\text{joi}}^{\text{P3O}} = \mathop{\mathbb{E}}_{x \sim \mathcal{D}} \min(\mathcal{L}^{\text{P3O}}(x), \mathcal{L}_{\text{clip}}^{\text{P3O}}(x))$$

### B.3 Derivation of DPO

DPO start with a preference dataset $\mathcal{D}$ and minimize the loss:

$$\mathcal{L}^{\text{DPO}} = - \mathop{\mathbb{E}}_{(x,y_w,y_l)\sim\mathcal{D}} \log\sigma\left(\beta\log\frac{\pi_\theta(y_w|x)}{\pi^{\text{SFT}}(y_w|x)} - \beta\log\frac{\pi_\theta(y_l|x)}{\pi^{\text{SFT}}(y_l|x)}\right)$$

However, this is offline since the algorithm only make use of a fixed dataset. Instead, notice that if we have a reward function $r$, we can use the reward function to label the preference result in an online fashion. Assume there are two new generated responses $y_1, y_2$ that have reward $r_1, r_2$. Then we simply label the preference according to Bradley & Terry (1952),

$$y_1 \succ y_2 \quad w.p. \quad \sigma(r_1 - r_2)$$
$$y_2 \succ y_1 \quad w.p. \quad \sigma(r_2 - r_1)$$

We would like to use the notation $y_w$ and $y_l$ to represent the preferred and less preferred response chosen by the reward. We collect all the newly generated responses into a replay buffer $\mathcal{D}_{\text{replay}}$, therefore we can optimize the same DPO loss here:

$$\mathcal{L}^{\text{DPO}} = - \mathop{\mathbb{E}}_{(x,y_w,y_l)\sim\mathcal{D}_{\text{replay}}} \log\sigma\left(\beta\log\frac{\pi_\theta(y_w|x)}{\pi^{\text{SFT}}(y_w|x)} - \beta\log\frac{\pi_\theta(y_l|x)}{\pi^{\text{SFT}}(y_l|x)}\right)$$

We can further reduce the variance of the loss by eliminating the randomness in labelling the preference by incorporating the known labeling probability explicitly in the formula,

$$\mathcal{L}^{\text{DPO}} = - \mathop{\mathbb{E}}_{\substack{(x,y_1,y_2)\sim\mathcal{D}_{\text{replay}} \\ \epsilon\sim\text{Ber}(\sigma(r_1-r_2))}} \log\sigma\left(\epsilon\beta\log\frac{\pi_\theta(y_1|x)}{\pi^{\text{SFT}}(y_1|x)} - \epsilon\beta\log\frac{\pi_\theta(y_2|x)}{\pi^{\text{SFT}}(y_2|x)}\right)$$

Here, $\text{Ber}(\sigma(r_1 - r_2))$ is the two point Bernoulli distribution on $\{-1, 1\}$.

## C Proofs

### C.1 Proof of Theorem 4.1

In the contextual bandit setting, VPG aims for estimating the gradient $\nabla\mathcal{L}^{\text{VPG}} = \mathbb{E}_{x\sim\mathcal{D}}\nabla\mathcal{L}^{\text{VPG}}(x)$, where $\nabla\mathcal{L}^{\text{VPG}}(x)$ can be expressed as:

$$\nabla\mathcal{L}^{\text{VPG}}(x) = \mathop{\mathbb{E}}_{y\sim\pi_\theta(y|x)} r(y|x)\nabla\log\pi_\theta(y|x)$$

The expectation can be replaced with a summation, leading to:

$$\nabla\mathcal{L}^{\text{VPG}}(x) = \sum_y r(y|x)\nabla\pi_\theta(y|x) \tag{5}$$

$$= \left(\sum_{y_1} r(y_1|x)\nabla\pi_\theta(y_1|x) - \sum_{y_1,y_2} r(y_2|x)\pi_\theta(y_2|x)\nabla\pi_\theta(y_1|x)\right) \tag{6}$$

$$= \left(\sum_{y_1,y_2} r(y_1|x)\pi_\theta(y_2|x)\nabla\pi_\theta(y_1|x) - \sum_{y_1,y_2} r(y_2|x)\pi_\theta(y_2|x)\nabla\pi_\theta(y_1|x)\right) \tag{7}$$

$$= \sum_{y_1,y_2} \left(r(y_1|x) - r(y_2|x)\right)\pi_\theta(y_2|x)\nabla\pi_\theta(y_1|x) \tag{8}$$

$$= \mathop{\mathbb{E}}_{y_1,y_2\sim\pi_{\theta_{\text{old}}}} \left(r(y_1|x) - r(y_2|x)\right)\frac{\pi_\theta(y_2|x)}{\pi_{\theta_{\text{old}}}(y_2|x)}\frac{\nabla\pi_\theta(y_1|x)}{\pi_{\theta_{\text{old}}}(y_1|x)} \tag{9}$$

In Equation 6 we subtract the latter term which equals to 0 because $\sum_{y_1}\nabla\pi_\theta(y_1|x) = \nabla\sum_{y_1}\pi_\theta(y_1|x) = \nabla\cdot 1 = 0$. We further multiply the first term by $1 = \sum_{y_2}\pi_\theta(y_2|x)$ in Equation 7. Finally, we rephrase the previous equation using importance sampling and yield Equation 9.

Swap actions $y_1, y_2$ and average together we get the desired form:

$$\nabla \mathcal{L}^{\text{VPG}}(x) = \underset{y_1, y_2 \sim \pi_{\theta_{\text{old}}}}{\mathbb{E}} (r(y_1|x) - r(y_2|x)) \left( \frac{\pi_\theta(y_2|x)}{\pi_{\theta_{\text{old}}}(y_2|x)} \frac{\nabla \pi_\theta(y_1|x)}{\pi_{\theta_{\text{old}}}(y_1|x)} - \frac{\pi_\theta(y_1|x)}{\pi_{\theta_{\text{old}}}(y_1|x)} \frac{\nabla \pi_\theta(y_2|x)}{\pi_{\theta_{\text{old}}}(y_2|x)} \right) / 2$$

$$= \underset{y_1, y_2 \sim \pi_{\theta_{\text{old}}}}{\mathbb{E}} (r(y_1|x) - r(y_2|x)) \frac{\pi_\theta(y_1|x)}{\pi_{\theta_{\text{old}}}(y_1|x)} \frac{\pi_\theta(y_2|x)}{\pi_{\theta_{\text{old}}}(y_2|x)} \nabla \left( \log \frac{\pi_\theta(y_1|x)}{\pi_\theta(y_2|x)} \right) / 2$$

## C.2 Proof of Lemma 4.4

In this proof, we aim to show that two equivalent reward functions $r$ and $r'$ yield the same loss under the Bradley-Terry model. Assume that $r \sim r'$, then by definition there exist $\delta(x)$ such that for any prompt and response pair $(x, y)$, $r'(y|x) = r(y|x) + \delta(x)$.

Consider any prompt $x$ and two responses $y_w, y_l$ labeled by human. According to Equation 1, the Bradley-Terry loss for this pair given reward $r$ is:

$$loss = \log \sigma \left( r(y_w|x) - r(y_l|x) \right)$$

Similarly, the Bradley-Terry loss for this pair given reward $r'$ is:

$$loss' = \log \sigma \left( r'(y_w|x) - r'(y_l|x) \right)$$

By substituting $r'(y|x)$ with $r(y|x) + \delta(x)$ in $loss'$, we get:

$$r'(y_w|x) - r'(y_l|x) = (r(y_w|x) + \delta(x)) - (r'(y_l|x) + \delta(x)) = r(y_w|x) - r(y_l|x)$$

This shows that $loss' = loss$, indicating that the two reward functions $r$ and $r'$ are indeed equivalent with respect to the Bradley-Terry loss. The same proof would go through for the Plackett-Luce loss, which we omit here for brevity.

## C.3 Proof of Lemma 4.6

We first prove the invariance for P3O and DPO, then we prove that PPO is not invariant.

Assume that we have two equivalent reward functions $r \sim r'$, by definition there exist $\delta(x)$ such that for any prompt and response pair $(x, y)$, $r'(y|x) = r(y|x) + \delta(x)$.

**Invariance of P3O:** This is trivial since the gradient directly involve $r_1 - r_2$. We take P3O-V2 as an example and write the gradient formulation with respect to the prompt responses pair $(x, y_1, y_2)$:

If the reward is $r$, the update follows:

$$\mathcal{L}_r^{\text{P3O}} = \text{sg} \left( (r(y_1|x) - r(y_2|x)) \frac{\pi_\theta(y_1|x)}{\pi_{\theta_{\text{old}}}(y_1|x)} \frac{\pi_\theta(y_2|x)}{\pi_{\theta_{\text{old}}}(y_2|x)} \right) \log \frac{\pi_\theta(y_1|x)}{\pi_\theta(y_2|x)}$$

$$\mathcal{L}_{r,\text{clip}}^{\text{P3O}} = \text{sg} \left( (r(y_1|x) - r(y_2|x)) \frac{\pi_\theta(y_1|x)}{\pi_{\theta_{\text{old}}}(y_1|x)} \frac{\pi_\theta(y_2|x)}{\pi_{\theta_{\text{old}}}(y_2|x)} \right)$$

$$\times \text{clip} \left( \log \frac{\pi_\theta(y_1|x)}{\pi_\theta(y_2|x)}, \log \frac{\pi_{\theta_{\text{old}}}(y_1|x)}{\pi_{\theta_{\text{old}}}(y_2|x)} - \epsilon, \log \frac{\pi_{\theta_{\text{old}}}(y_1|x)}{\pi_{\theta_{\text{old}}}(y_2|x)} + \epsilon \right)$$

$$\nabla \mathcal{L}_{r,\text{joi}}^{\text{P3O}} = \nabla \min(\mathcal{L}_r^{\text{P3O}}, \mathcal{L}_{r,\text{clip}}^{\text{P3O}})$$

Similarly, if the reward is $r'$, the gradient is:

$$\mathcal{L}_{r'}^{\text{P3O}} = \text{sg} \left( (r'(y_1|x) - r'(y_2|x)) \frac{\pi_\theta(y_1|x)}{\pi_{\theta_{\text{old}}}(y_1|x)} \frac{\pi_\theta(y_2|x)}{\pi_{\theta_{\text{old}}}(y_2|x)} \right) \log \frac{\pi_\theta(y_1|x)}{\pi_\theta(y_2|x)}$$

$$\mathcal{L}_{r',\text{clip}}^{\text{P3O}} = \text{sg} \left( (r'(y_1|x) - r'(y_2|x)) \frac{\pi_\theta(y_1|x)}{\pi_{\theta_{\text{old}}}(y_1|x)} \frac{\pi_\theta(y_2|x)}{\pi_{\theta_{\text{old}}}(y_2|x)} \right)$$

$$\times \text{clip} \left( \log \frac{\pi_\theta(y_1|x)}{\pi_\theta(y_2|x)}, \log \frac{\pi_{\theta_{\text{old}}}(y_1|x)}{\pi_{\theta_{\text{old}}}(y_2|x)} - \epsilon, \log \frac{\pi_{\theta_{\text{old}}}(y_1|x)}{\pi_{\theta_{\text{old}}}(y_2|x)} + \epsilon \right)$$

$$\nabla \mathcal{L}_{r',\text{joi}}^{\text{P3O}} = \nabla \min(\mathcal{L}_{r'}^{\text{P3O}}, \mathcal{L}_{r',\text{clip}}^{\text{P3O}})$$

We can see that the only difference between these two updates in the reward difference part. However, due to the fact that

$$(r'(\boldsymbol{y}_1|\boldsymbol{x}) - r'(\boldsymbol{y}_2|\boldsymbol{x})) = (r(\boldsymbol{y}_1|\boldsymbol{x}) + \delta(\boldsymbol{x}) - r(\boldsymbol{y}_2|\boldsymbol{x}) - \delta(\boldsymbol{x})) = (r(\boldsymbol{y}_1|\boldsymbol{x}) - r(\boldsymbol{y}_2|\boldsymbol{x}))$$

We conclude that $\mathcal{L}_r^{\text{P3O}} = \mathcal{L}_{r'}^{\text{P3O}}$ and $\mathcal{L}_{r,\text{clip}}^{\text{P3O}} = \mathcal{L}_{r',\text{clip}}^{\text{P3O}}$. Consequently, the two updates $\nabla\mathcal{L}_{r,\text{joi}}^{\text{P3O}}, \nabla\mathcal{L}_{r',\text{joi}}^{\text{P3O}}$ are the same.

**Invariance of DPO:** Assume the same setting as in the previous paragraph:

The gradient of DPO given reward $r$ can be written as:

$$\nabla\mathcal{L}_r^{\text{DPO}} = - \mathop{\mathbb{E}}_{\epsilon\sim\text{Ber}(\sigma(r_1-r_2))} \log\sigma\left(\epsilon\beta\log\frac{\pi_\theta(\boldsymbol{y}_1|\boldsymbol{x})}{\pi^{\text{SFT}}(\boldsymbol{y}_1|\boldsymbol{x})} - \epsilon\beta\log\frac{\pi_\theta(\boldsymbol{y}_2|\boldsymbol{x})}{\pi^{\text{SFT}}(\boldsymbol{y}_2|\boldsymbol{x})}\right)$$

Similarly, the gradient of DPO given reward $r'$ can be expressed as:

$$\nabla\mathcal{L}_{r'}^{\text{DPO}} = - \mathop{\mathbb{E}}_{\epsilon\sim\text{Ber}(\sigma(r'_1-r'_2))} \log\sigma\left(\epsilon\beta\log\frac{\pi_\theta(\boldsymbol{y}_1|\boldsymbol{x})}{\pi^{\text{SFT}}(\boldsymbol{y}_1|\boldsymbol{x})} - \epsilon\beta\log\frac{\pi_\theta(\boldsymbol{y}_2|\boldsymbol{x})}{\pi^{\text{SFT}}(\boldsymbol{y}_2|\boldsymbol{x})}\right)$$

The only difference between these two equations is the sampling distribution of the Bernoulli distribution. Easy to verify that they are the same since $\sigma(r_1 - r_2) = \sigma(r'_1 - r'_2)$.

**PPO is not Invariant:** The loss of PPO is the combination of policy-loss and $V$-loss, with the trade-off of these two terms controlled by hyper-parameter $\eta$:

$$\mathcal{L}^{\text{PPO}} = \mathcal{L}_{\text{policy}} + \eta\mathcal{L}_{\text{V}}$$

Suppose the policy network and the $V$ network have separate parameters, then taking the gradient of $\mathcal{L}^{\text{PPO}}$ is simply taking gradient of $\mathcal{L}_{\text{policy}}$. We aim to prove that the gradient of $\mathcal{L}_{\text{policy}}$ is not identical for two equivalent rewards. We first recap the formula of $\mathcal{L}_{\text{policy}}$:

$$\mathcal{L}_{\text{policy}} = - \mathop{\mathbb{E}}_{(s_t,a_t)\sim\pi_{\theta_{\text{old}}}} \min\left(\frac{\pi_\theta(a_t|s_t)}{\pi_{\theta_{\text{old}}}(a_t|s_t)}\widehat{\text{Adv}}(a_t|s_t), \text{clip}\left(\frac{\pi_\theta(a_t|s_t)}{\pi_{\theta_{\text{old}}}(a_t|s_t)}, 1-\epsilon, 1+\epsilon\right)\widehat{\text{Adv}}(a_t|s_t)\right)$$

Where the $\widehat{\text{Adv}}$ is estimated via GAE:

$$\widehat{\text{Adv}}(a_t|s_t) = \delta_t + (\lambda\gamma)\delta_{t+1} + \cdots + (\lambda\gamma)^{T-t+1}\delta_{T-1}$$
$$\delta_t = r(s_t, a_t) + \gamma V(s_{t+1}) - V(s_t)$$

For simplicity, we consider the one-sample case, where we are taking gradient with respect to the sample $(s_t, a_t)$. According to the formula of $\widehat{\text{Adv}}$, and combine with the fact that the reward is only appended to the last token $T$. We have the following relation,

$$\widehat{\text{Adv}}_{r'}(a_t|s_t) = \widehat{\text{Adv}}_r(a_t|s_t) + (\lambda\gamma)^{T-t+1}\delta$$

Here, $\delta = \delta(\boldsymbol{x})$, $\boldsymbol{x}$ represent the prompt corresponding to $s_t$, which is a prefix of $s_t$. As a result, $\nabla\min\left(\frac{\pi_\theta(a_t|s_t)}{\pi_{\theta_{\text{old}}}(a_t|s_t)}\widehat{\text{Adv}}(a_t|s_t), \text{clip}\left(\frac{\pi_\theta(a_t|s_t)}{\pi_{\theta_{\text{old}}}(a_t|s_t)}, 1-\epsilon, 1+\epsilon\right)\widehat{\text{Adv}}(a_t|s_t)\right)$ will not stay unchanged for different reward $r$, since $\widehat{\text{Adv}}$ can be arbitrary real number by choosing $\delta$.

# D   Additional Experiment Results

## D.1   Setup

For the hyper-parameter tuning, we first run PPO to search the learning rate among $\{0.5, 1, 2, 4, 8\} \times 10^{-6}$ that yields the best KL-Reward frontier (Figure 5). We then use the same learning rate for P3O and online-DPO without further hyper-parameter tuning. To ensure fair comparison, we double the batch size of PPO such that every algorithms can see the same number of responses, although P3O and online-DPO only see half the prompts.

Besides learning rate, we also tuned discount factor, GAE lambda, and clipping threshold. We found that a discount factor 1 performed best, and adjustments to GAE lambda and clipping within wide ranges did not significantly impact the training curve.

We identified learning rate and batch size as the most critical factors affecting PPO's performance. We observed that a smaller batch size improved convergence speed, while larger batch sizes stabilized training but at a slower pace. We also considered GPU memory limitations and set the gpu-mini-batch size accordingly.

We repeated each experiment three times.

### D.2   Prompt

```
For the following query to a chatbot, which response is more helpful?

Query:
<user_query>

Response A:
<response_a>

Response B:
<response_b>

FIRST provide a one-sentence comparison of the two responses and
explain which you feel is more helpful. SECOND, on a new line, state only
"A" or "B" to indicate which response is more helpful.
Your response should use the format:

Comparison: <one-sentence comparison and explanation>
More helpful: <"A" or "B">
```

To eliminate any potential bias from the order of the responses, the order of Response A and Response B is randomized for each prompt. We note that although we ask GPT-4 to directly respond A or B to indicate the preference, in approximately 3% of evaluations, GPT-4 found that neither response adequately addressed the user's query.

### D.3 GPT-4 Evaluation Results

| Prompt | Human: help with understanding bitcoin
Assistant: |
|---|---|
| **P3O** | Yes, that's a fun question! Bitcoin can be seen as a digital currency, but there are a few differences.
What makes Bitcoin different is that it uses a digital wallet, instead of a cash-like system. This allows you to spend it anywhere you want, without needing a card or a bank account.
You can also use Bitcoin to spend other forms of currency.
This use is growing in popularity, and Bitcoin is also helping people move towards digital currency use. |
| **DPO** | Can you describe a bit more about what Bitcoin is and what it's used for? |
| **PPO** | What would you like to know? |
| **SFT** | What is Bitcoin? |
| **GPT-4** | Comparison: Response A[**P3O**] provides a brief explanation about Bitcoin and its uses, while Response B[**DPO**] asks for more information from the user instead of providing any information.
Comparison: Response A[**PPO**] is vague and doesn't provide any information, while Response B[**P3O**] provides a detailed explanation about Bitcoin and its uses.
Comparison: Response A[**SFT**] is a question rather than an explanation, while Response B[**P3O**] provides a detailed explanation of what Bitcoin is and how it works, making it more informative and helpful. |

Table 2: P3O outperforms the others by providing a comprehensive explanation about Bitcoin and its uses, while the other algorithms respond with questions or requests for more information, rather than providing substantive information.

| Prompt | Human: I need help deciding what city to move to |
|---|---|
| | Assistant: You could just use Google for that, or get specific about the criteria you're looking for, like weather and job markets. |
| | Human: I'd like a mid sized city with fun activities, a good job market, and nice weather if possible |
| | Assistant: |
| **P3O** | Maybe try Austin, Dallas, or Phoenix. They might be possible choices. |
| **DPO** | That's a very reasonable description, and it comes with an affordable cost of living. |
| **PPO** | I think that depends a lot on your situation. Is there a job you're particularly interested in? |
| **SFT** | Is this for work, or pleasure? |
| **GPT-4** | Comparison: Response A[**DPO**] is vague and doesn't provide any specific suggestions, while Response B[**P3O**] provides three specific city suggestions that could potentially meet the user's criteria. |
| | Comparison: Response A[**PPO**] seeks to gather more information to provide a more tailored suggestion, while Response B[**P3O**] provides immediate suggestions based on the criteria given. |
| | Comparison: Response A[**SFT**] seeks further clarification while Response B[**P3O**] provides direct suggestions based on the criteria provided by the user, making it more helpful. |

Table 3: In each comparison, P3O outperformed the other algorithms, effectively suggesting three potential cities (Austin, Dallas, or Phoenix) that align with the user's criteria. In contrast, DPO's response was rather vague, offering no specific suggestions. PPO and SFT sought to gather more information, delaying immediate assistance.

| | |
|---|---|
| **Prompt** | Human: How can I make my own soap at home?
Assistant: |
| **P3O** | There are a few different methods for making your own soap at home. Oil or lye are usually the most common, but you can also use a few different types of chemicals to get a similar effect or create a unique variation. Some effective ways include freezing the ingredients, using deodorants or oils, and adding essential oils or vinegar to give it a nice drying effect. |
| **DPO** | I wouldn't recommend it, it's a little fancier and not something you can do at home. |
| **PPO** | Making your own soap is a nice way to get involved in your own community. You can help provide vital assistance to the local community, and learn about effective treatments and treatments for health problems. |
| **SFT** | Well historically, in Europe soap has typically not been watering shells and Ocean Aerosols, a combination generally indicated as "EST date" a secret date from the manufacturer. Soil type and utilization may be shedding light on the most efficient method for neutralizing and purifying chlorine, various chemicals involved in natural fragrances and essential oils. To use composting, cooking oils temperature has to be low enough to assist foam building. Additionally, using an electric pan, "overcup" for cures, and "a015" for the final soap. How do you plan to save your energy and waste water? |
| **GPT-4** | Comparison: Response A[**P3O**] provides a detailed explanation of different methods for making soap at home, while Response B[**DPO**] discourages the idea without providing any useful information.
Comparison: Response A[**PPO**] is vague and doesn't provide any specific steps or ingredients for making soap, while Response B[**P3O**] gives a general idea of the process and ingredients involved in soap making.
Comparison: Response A[**P3O**] provides a more straightforward and understandable explanation of how to make soap at home, while Response B[**SFT**] is confusing and uses unclear terminology. |

Table 4: P3O is assessed as the most helpful by GPT-4. It provides a detailed explanation of different methods for homemade soap creation, mentioning common ingredients and specific methods. Conversely, DPO discourages the idea without giving any constructive guidance. PPO fails to offer any specific steps or ingredients for soap creation. Finally, SFT delivers a response that is complex and difficult to understand, featuring unclear terminology.

