# OpenReview forum: "Pairwise Proximal Policy Optimization: Language Model Alignment with Comparative RL"
_colmweb.org/COLM/2024/Conference — COLM_

### Official Review · Reviewer_cNS6 · 2024-04-28

**Rating:** 7
**Confidence:** 3
**Ethics Flag:** 1

**Summary:**

The paper presents a new RL-based LLM alignment method called P3O and demonstrates its effectiveness with a few small scale RMs and Policy Models on two datasets. The authors claim the traditional PPO algorithm may produce varying updates even given the identical human preference data and P3O algorithm is a good remedy for this issue. Both theoretical and empirical results are presented to show that P3O can more effectively align with human preferences than existing methods, suggesting that comparative feedback could be a superior method for training LLMs to adhere to human values.

**Reasons To Accept:**

1. The paper studies an important problem and presents a principled solution based on some in-depth analysis of existing methods

2. The experiments, although with limited scope, are still pretty solid

**Reasons To Reject:**

I don't see any major issues but I do feel a few places can be improved/clarified.

First, I feel the overall paper writing/presentation is not very good. Some parts of the paper have too dense information. For example, the page 4 section 4.1 is very dense and more discussions are needed. The bad presentation makes the some insights of this paper inaccessible. For example, I feel the reward equivalence in definition 4.3 is very interesting and should be further emphasize. The current way of presentation makes me feel it is defined in a way that can make lemma 4.4 hold (plus make the DPO/P3O enjoys the invariance property while the PPO validates this property). More intuitive explanations on why define the reward equivalence are needed.

Second, I think the experiments can be further improved in a few ways. First, you can consider more mid-sized (10B-70B) LLMs and show P3O method can generalize. Second, I think more comparison with other online version of DPO-style algorithms (e.g., SLiC [1], RSO [2]) are useful. You current experiment seems already include the online DPO (the end of page 7), why not check other online improved DPO-style algorithms? Finally, I feel the connection between your theory and experiments are not very strong. Your empirical results show P3O is better than PPO/DPO but I don't know if those improvements are due to your claimed theoretical reasons (e.g., resolving RM training, RL finetuning inconsistency, addressing invariant issue, etc) or maybe just come from less implementation noises. I believe some ablation studies to better bridge theoretical insights and empirical results are helpful.

[1]. SLiC-HF: Sequence Likelihood Calibration with Human Feedback

[2]. Statistical Rejection Sampling Improves Preference Optimization

---

> ### Author Rebuttal · Authors · 2024-05-31
>
> We appreciate the reviewer's feedback and recognize the importance of improving the overall presentation of our paper.
> >…overall paper writing/presentation is not very good…
>
>
> We will provide a clearer explanation of the reward equivalence definition and its significance. We will include additional derivations and discussions of our algorithm in the appendix.
>
>
> >More intuitive explanations on why define the reward equivalence are needed.
>
>
> The definition of reward equivalence is motivated by the observation that different prompts can yield significantly varying reward values. Suppose we have two prompts, one being relatively easy and the other more challenging. For the easy prompt, a standard response is likely to receive a high reward. In contrast, even a more sophisticated response to the hard prompt may result in a lower reward.
>
> Methods that rely solely on absolute rewards, such as PPO, tend to favor increasing the likelihood of the easy prompt response while decreasing the likelihood for the hard one. This can be detrimental to the model's performance.
>
> P3O circumvents this issue by utilizing comparative signals. By focusing on the relative preferences between responses, P3O is not sensitive to the absolute reward values associated with different prompts.
>
> We will provide additional discussions to further clarify the motivation behind the reward equivalence definition and the advantages it brings to P3O.
>
>
> >…experiments can be further improved…
>
>
> The bottleneck of tuning larger models (7b+) is on the infrastructure side, where there’s some conflict between ZeRO3 and generation. For the baseline, we only choose DPO as we think it's simple and well adopted. We’ll compare with SLiC or RSO if time permits.
>
> >…connection between your theory and experiments are not very strong…
>
> The reward shift invariant issue pointed out by our theoretical analysis significantly impacts the empirical performance of PPO. Our observations reveal that adding a constant to the entire reward signal can dramatically alter the model's behavior after tuning with PPO. If the reward is generally negative, the model tends to generate excessively long responses, while a positive reward conversely leads to significantly shorter outputs. In practice, precisely balancing such constant shifts is crucial for PPO. In contrast, P3O consistently produces identical results regardless of constant shifts in the reward, thereby eliminating a significant source of instability inherent to PPO.

---

> > ### Comment · Reviewer_cNS6 · 2024-06-05
> >
> > Thank you for your response. I've read it and my assessment remains the same.

---

### Official Review · Reviewer_8nNu · 2024-05-09

**Rating:** 6
**Confidence:** 4
**Ethics Flag:** 1

**Summary:**

The submission proposes a pairwise PPO method to improve PPO in online LM alignment process. The major argument is the gap between reward model learning from pairwise preference and RLHF with the pointwise policy gradient process.

*Novelty/Forward-outlook*
DPO is also using the proposed (general) BT-based loss. Other methods such as SliC use pairwise losses that mitigate the constant shift issue. There’re also preference-based RL methods in the literature. As acknowledged by the authors, the work can be seen as combining PPO and DPO. Thus, the proposed method is specific to improving the online PPO and the broader technical impact is reasonable but not substantial to the reviewer.

The authors may consider discussing a recent work [1], which unified several pairwise objectives under the learning to rank literature. It did not cover the Pairwise PPO method in this paper since it is not used in the learning to rank literature, so the difference here might be an interesting way to highlight the novelty of this submission.

*Depth/Technical impact*
Despite the comparisons between online vs offline alignment and popularity of methods such as DPO, given the current popularity of PPO, the contribution is reasonable to the reviewer, but not substantial.

The scale invariant problem is well studied in the learning to rank literature [2]. Thus the invariance analysis is interesting to the field but not novel, and it might be good to discuss such connections to related fields.

*Experimental evaluation*
Experiments are conducted on Reddit Tl;Dr and Anthropic HH, which are commonly used in the field. Evaluation metrics are proxy reward and GPT-4 auto side by side. Baselines are DPO and PPO. These settings are borderline and are typically minimally required for papers in the field. Extra steps include human eval, more datasets, more baselines, more ablation studies and model/size variants of the policy / reward model.

The reviewer is also not convinced about the experimental fairness. It is well-known that methods such as PPO have huge variance. The reviewer did not find convincing discussions about the model tuning and selection process except for brief tuning of learning rate in Section D.1, and one instance from each method is analyzed. It would be more convincing to see deeper analysis on the variance behaviors to reduce doubts about cherry picking.

*Clarity*
The paper is clear to the reviewer.

*Overall evaluation*
Overall the paper is borderline to the reviewer. The basic idea makes a meaningful contribution to the field, however its impact may not substantially wide given existing work in the field and related fields. The theoretical analysis has value but not substantial. The experimental validation only meets the minimal requirements of papers in the field and does not make a strong case.

[1] LiPO: Listwise Preference Optimization through Learning-to-Rank. Arxiv 2024
[2] Scale Calibration of Deep Ranking Models. KDD 2022

**Reasons To Accept:**

Below are from the overall review, please refer to the overall review for details.

- The proposed method is specific to improving the online PPO and the broader technical impact is reasonable given the popularity of PPO.

- The theoretical analysis looks correct and makes contribution to the important field.

**Reasons To Reject:**

Below are from the overall review, please refer to the overall review for details.

- Pairwise methods similar to this and invariance analysis are done in the literature in related fields such as RL and learning to rank.

- Evaluations are not very convincing and only meets the minimal requirements of papers in this field. Risk of cherry-picking.

---

> ### Author Rebuttal · Authors · 2024-05-31
>
> We appreciate the reviewer's insightful comments.
> >Novelty/Forward-outlook…
>
>
> In the final draft, we will discuss preference-based RL and learning to rank literature. We note that P3O shares similarities with DPO and LiPO in that they all utilize comparative feedback. However, a key distinction lies in the fact that P3O is a policy gradient algorithm, benefiting from its strict improvement property. This sets P3O apart from ranking loss-based methods like DPO.
>
>
> >Depth/Technical impact…
>
>
> We will provide a comprehensive discussion of the scale invariance issue, including its relevance to the learning-to-rank literature [2].
>
>
> By explicitly studying the scale invariance problem and its impact on training stability and sample efficiency, we bridge the gap between the alignment community and insights from other fields.
>
>
> >Experimental evaluation…
>
>
>
>
> Our choice of baselines, and evaluation metrics are standard and widely accepted in the field, following the precedent works [RAFT APA DPO].
> Evaluation metrics: We have included the GPT-4 side-by-side evaluation, which has been shown to correlate strongly with human judgments [Rafailov et al., 2023]. We acknowledge the importance of human evaluation and will consider it for future work if resources allow.
> Baselines: We have compared our method with two strong baselines, DPO and PPO, which are established algorithms in the field.
> Regarding ablations, we have provided an analysis of the effects of clipping and KL constraint in the appendix.
>
> >...not convinced about the experimental fairness…
>
> We performed extensive hyperparameter tuning, and we apologize if these details were not clear:
>
>
> **Hyperparameter analysis**: We conducted a comprehensive analysis of hyperparameters for PPO, including discount factor, GAE lambda, and clipping threshold. We found that a discount factor 1 performed best, and adjustments to GAE lambda and clipping within wide ranges did not significantly impact the training curve.
>
> **lr and batch size**: We identified lr and batch size as the most critical factors affecting PPO's performance. We observed that a smaller batch size improved convergence speed, while larger batch sizes stabilized training but at a slower pace. We also considered GPU memory limitations and potential gradient accumulation bugs and set the gpu-mini-batch size accordingly.
> Seed selection: We repeated each experiment three times.
>
>
> We believe that this additional information will address the reviewer's concerns.

---

> > ### Comment · Reviewer_8nNu · 2024-06-04
> >
> > Thanks for the response. I think they are reasonable and will keep the score.

---

### Official Review · Reviewer_qYdp · 2024-05-15

**Rating:** 7
**Confidence:** 3
**Ethics Flag:** 1

**Summary:**

This paper proposes a simple and effective policy gradient algorithm called Pairwise Proximal Policy Optimization (P3O) to deal with the inconsistency between reward learning and RL in PPO. This algorithm employs an additional sample to estimate the gradient unbiasedly, and update the policy based on the comparison difference. P3O is invariant with any reward functions that contain identical preference information, and does not require learning a value function. Experimental results on summarization and question answering show the effectiveness of the proposed method compared with PPO and DPO.

**Reasons To Accept:**

1. The proposed method based on comparative feedback is novel and well-motivated. The derivation of P3O in Section 4 is clearly presented and easy to follow.
2. Experimental results show the superior performance of P3O over PPO and DPO, which indicate the potential impact of the proposed method on different LLM alignment tasks.
3. This paper is overall well-written. I enjoy reading this paper from the method to experiments.

**Reasons To Reject:**

1. The authors mention the instability issue of PPO in the abstract and say that the inconsistency between reward learning and RL stages exacerbates the instability. I wonder whether P3O with the ability to solve this inconsistency issue can be more stable during training than PPO. It would be better to provide the curve of reward / KL vs. training steps in the experiment.

2. The training efficiency should be also analyzed because the proposed method seems to use 2x generated samples to acquire a more stable estimation of gradients at each step compared with PPO. From my understanding, there exists a trade-off between efficiency and stability, but the current paper lacks the related analysis.

3. Since GPT-4 evaluation may have some biases according to existing works [1], I recommend the authors to add human evaluation to make the experimental results more convincing.

[1] Large Language Models are not Fair Evaluators.

---

> ### Author Rebuttal · Authors · 2024-05-31
>
> Thank you for your feedback. We appreciate the opportunity to enhance the clarity of our work.
> >...whether P3O with the ability to solve this inconsistency issue…
>
> We will incorporate a training curve to highlight the instability observed in PPO. Furthermore, we will present a comparative analysis of the reward / KL as a function of training steps for different methods in the appendix for better understanding.
> >...the proposed method seems to use 2x generated samples to acquire a more stable estimation of gradients at each step compared with PPO…
>
>
> To ensure a fair comparison between P3O and PPO, we have adopted a setting that potentially disadvantages P3O. Specifically, we use half the batch size for P3O compared to PPO. This adjustment guarantees that each gradient update across both methods is based on the same number of model-generated responses. However, it is important to note that under this setup, PPO benefits from processing a larger number of prompts per update, a factor generally considered advantageous for learning.
>
>
> Regarding hyperparameter tuning, we have conducted extensive experiments on various parameters for PPO, including Generalized Advantage Estimation lambda, discount factor, batch size, and learning rate. We have identified the optimal configuration for PPO and adapted it to be compatible with P3O without conducting additional hyperparameter searches.
>
> >...add human evaluation to make the experimental results more convincing.
>
> We chose GPT-4 evaluation for two reasons: its efficiency in approximating human preference and its reproducibility. Studies such as [Rafailov et al., 2023] have shown that GPT-4 judgments correlate strongly with human evaluations, with inter-annotator agreement often similar or higher than human agreement. Additionally, [Starling 7b] demonstrated significant improvement in an aligned model using GPT-4 judgments and RLHF on the live human evaluation platform like Chatbot Arena.
>
>
> We recognize that GPT-4 evaluation has advantages in terms of efficiency and reproducibility, as labeller biases are difficult to avoid and document. The usage of GPT4 will allow us to directly compare the results obtained from GPT-4 evaluation and human evaluation, providing a more robust assessment of our proposed method's performance.
>
>
> We will also provide more responses in the appendix. This will offer additional insights for curious readers and further validate the effectiveness of our proposed approach.

---

> > ### Comment · Reviewer_qYdp · 2024-06-05
> >
> > Thanks for your response. I have read the rebuttal which solves most of my concerns. I decide to maintain my score.

---

### Official Review · Reviewer_uGsp · 2024-05-23

**Rating:** 6
**Confidence:** 4
**Ethics Flag:** 1

**Summary:**

This paper proposes a new algorithm for RLHF, called Pairwise Proximal Policy Optimization (P3O). In an online training fashion similar to PPO, this algorithm uses a comparative signal between two sampled responses to train the policy model, which aligns better with how the reward model is trained. It also demonstrates some other properties, such as removing the need for a value model and being invariant to "information-equivalent" reward functions. The author provides detailed derivations of the P3O algorithm based off vanilla Policy Gradient, and also conducts experiments on summarization and question answering tasks to show the superiority of P3O over PPO and DPO, in terms of KL-Reward frontier and GPT4-judged win-rate.

**Questions To Authors:**

In Theorem 4, you introduce the importance sampling trick and claim it is an unbiased estimation of the pairwise policy gradient loss. I don't quite understand the intuition of this trick. Specifically, why do you choose $\theta_{old}$ to compare against? Why is it called importance sampling? And why is it unbiased? Did you observe the difference between adding this term versus not adding it?

**Reasons To Accept:**

- The newly proposed algorithm is well motivated by the discrepancy between reward model training loss and how they are commonly used in typical PPO algorithms.
The author provides a detailed derivation from Policy Gradient to P3O and discusses their algorithm compared to PPO and DPO, which I enjoyed reading. The use of pairwise policy gradient for preference-based learning looks creative and sound to me.
- The results align well with the author's claims, showing the effectiveness of the proposed model.

**Reasons To Reject:**

- The derivations may have some flaws that I want the author to clarify. I will increase the score if I can be convinced.
- I feel that the experiments are relatively weak in terms of:
   - They mainly use some relatively old and weak pretrained models, including gpt-j 6B and pythia 1B or 6B. I don't understand why not switching to LLaMa 7B if the 6B model training is feasible. Using more recent models can also strengthen the argument that this algorithm can be applied to stronger LMs.
   - The chosen tasks are reasonable. However, I feel that just showing the KL-reward frontier is not enough, as the reward models may be biased by some superficial features, and the KL-reward frontier may just be comparing how strong the algorithms are learning such unwanted features within the KL budget. Similarly, GPT4-based evaluation also has some superficial features. I think the experiments can be strengthened by incorporating some reliable metrics, or including some tasks (e.g., math) that can be easily verifiable, or some human evaluation.

---

> ### Author Rebuttal · Authors · 2024-05-31
>
> Thank you for your feedback. We appreciate your recognition of the novelty of our paper. We will endeavor to address your concerns in our revised draft.
> >They mainly use some relatively old and weak pretrained models…
>
> The decision to employ these specific models was driven by considerations of flexibility and consistency. The Pythia series offers a wide range of model sizes, from 70M to 12B parameters. This flexibility in sizes is particularly valuable in research settings, where we’re able to initiate our experiments with smaller, more manageable 1B models on V100s and then scale up to larger 6B models on A100s. Furthermore, employing the pythia models ensures a consistent framework for evaluation across different model sizes, with minimal code changing.
> >...experiments can be strengthened by incorporating some reliable metrics…
>
> We have incorporated GPT-4 evaluation into our methodology, primarily due to its cost-effectiveness and the strong correlation with human preferences they have demonstrated. For example, (Rafailov et al., 2023) provides compelling evidence of GPT-4 judgments' alignment with human evaluations, noting that agreement between GPT-4 and human annotators is often on par with or exceeds that among humans themselves. Furthermore, the study by (Starling-7b) highlights the effectiveness of using GPT-4 feedback for Reinforcement Learning from Human Feedback (RLHF), where models trained on GPT-4 judgments showed significant improvements in performance on the live human evaluation platform, Chatbot Arena. As a complementary, we will add more interesting responses and compare different methods in the appendix, which may be interesting for curious reviewers.
> >...importance sampling trick…
>
>
> Importance sampling is a crucial technique in reinforcement learning, particularly when dealing with the challenge of updating a policy 𝜋 based on data collected from an older (hence different) policy 𝜋_old. This is common in RL as we usually collect a batch of samples and perform multiple gradient updates. The term "importance sampling" refers to this process of reweighting the contributions of each sample to reduce bias. Specifically, the importance sampling ratio π(a|s) / π_old(a|s) adjusts the influence of each experience based on how much more or less likely it is under the current policy compared to the old policy. This makes the estimator unbiased. In practice, we find the correction to be helpful when combined with clipping.

---

### Decision · Program_Chairs · 2024-07-10

**Decision:**

Accept

**Comment:**

For alignment purposes the most common data that is gathered is preference-based: given `n` completions (mostly `n==2`) annotators rank them according to their preference and a reward model is learnt from that. This is then used in a RL learning with a reward model as proxy, but the RL algorithm used does not integrate explicitly the pair-wise comparison.

This paper starts from this discrepancy, and proposes to use the comparative signal to train the policy model. This is well motivated, cleanly derivated and experimentally proven to improve the final model quality.
While one reviewer found some parts of the explanation lacking, most praised the clarity of the writing (the verb "to enjoy" being used twice)

The reviewers pointed out several additional experiments which would enrich the paper and clarify some parts, and I hope that the authors will fulfil their commitment of integrating those